# Novel Purine Derivative ITH15004 Facilitates Exocytosis through a Mitochondrial Calcium-Mediated Mechanism

**DOI:** 10.3390/ijms23010440

**Published:** 2021-12-31

**Authors:** Ricardo de Pascual, Francesco Calzaferri, Paula C. Gonzalo, Rubén Serrano-Nieto, Cristóbal de los Ríos, Antonio G. García, Luis Gandía

**Affiliations:** 1Instituto Teófilo Hernando and Departamento de Farmacología y Terapéutica, Universidad Autónoma de Madrid, C/Arzobispo Morcillo, 4, 28029 Madrid, Spain; ricardo.pascual@uam.es (R.d.P.); francesco.calzaferri@gmail.com (F.C.); pauladgo@ucm.es (P.C.G.); ruben.serrano@imdea.org (R.S.-N.); cristobal.delosrios@urjc.es (C.d.l.R.); antonio.garcia@ifth.es (A.G.G.); 2Instituto de Investigación Sanitaria, Hospital Universitario de la Princesa, C/Diego de León, 62, 28006 Madrid, Spain; 3Institut des Biomolécules Max Mousseron (IBMM—UMR5247, CNRS), 1919 Route de Mende, CEDEX 5, 34293 Montpellier, France; 4Departamento de Ciencias Básicas de la Salud, Campus de Alcorcon, Universidad Rey Juan Carlos, Avda. Atenas s/n, 28922 Alcorcón, Spain; 5Fundación Teófilo Hernando, Parque Científico de Madrid, Campus de Cantoblanco, 28049 Madrid, Spain

**Keywords:** mitochondria, ITH15004, chromaffin cell, catecholamine release, calcium signalling, neurodegeneration

## Abstract

Upon depolarization of chromaffin cells (CCs), a prompt release of catecholamines occurs. This event is triggered by a subplasmalemmal high-Ca^2+^ microdomain (HCMD) generated by Ca^2+^ entry through nearby voltage-activated calcium channels. HCMD is efficiently cleared by local mitochondria that avidly take up Ca^2+^ through their uniporter (MICU), then released back to the cytosol through mitochondrial Na^+^/Ca^2+^ exchanger (MNCX). We found that newly synthesized derivative ITH15004 facilitated the release of catecholamines triggered from high K^+^-depolarized bovine CCs. Such effect seemed to be due to regulation of mitochondrial Ca^2+^ circulation because: (i) FCCP-potentiated secretory responses decay was prevented by ITH15004; (ii) combination of FCCP and ITH15004 exerted additive secretion potentiation; (iii) such additive potentiation was dissipated by the MICU blocker ruthenium red (RR) or the MNCX blocker CGP37157 (CGP); (iv) combination of FCCP and ITH15004 produced both additive augmentation of cytosolic Ca^2+^ concentrations ([Ca^2+^]_c_) K^+^-challenged BCCs, and (v) non-inactivated [Ca^2+^]_c_ transient when exposed to RR or CGP. On pharmacological grounds, data suggest that ITH15004 facilitates exocytosis by acting on mitochondria-controlled Ca^2+^ handling during K^+^ depolarization. These observations clearly show that ITH15004 is a novel pharmacological tool to study the role of mitochondria in the regulation of the bioenergetics and exocytosis in excitable cells.

## 1. Introduction

By acting on receptors, ion channels or intracellular signalling pathways, various endogenous neurotransmitters and exogenous compounds have been implicated in the positive or negative regulation of the exocytotic release of catecholamines from adrenal medulla chromaffin cells (CCs) [1]. In this physiological process, Ca^2+^ ions play a central role in triggering the stimulus-secretion coupling process [2]. During CC activity, cytosolic Ca^2+^ concentrations ([Ca^2+^]_c_) may undergo brisk sudden changes from its basal concentrations of around 0.1 µM to tens micromolar at exocytosis subplasmalemmal sites, giving rise to vesicle fusion and fast exocytosis. Milder 0.5–1 µM [Ca^2+^]_c_ elevations, as a result of Ca^2+^ redistribution, occur at inner cytosolic areas to regulate pre-exocytotic steps, such as Ca^2+^-dependent vesicle transport to reload the secretory machinery [1,3,4,5].

Those Ca^2+^ microdomains (CMDs) are originated and tightly regulated by endogenous messengers including Ca^2+^ itself and various pharmacological ligands acting on Ca^2+^ entry through voltage-activated calcium channels (VACCs), their redistribution into the endoplasmic reticulum (ER) Ca^2+^ store and mitochondria, and its subsequent release into the cytosol from those CC organelles. For instance, by stimulating the release of ER Ca^2+^ in bovine CCs (BCCs), we have previously shown that the Na^+^,K^+^-ATPase pump blocker ouabain [6] and, more recently, the vesicle dopamine transporter blocker tetrabenazine (TBZ) [7] facilitated the depolarizing-evoked catecholamine release through the mobilization of ER Ca^2+^. These findings could have interest in the context of safety issues with the handling of these drugs in clinical sets (i.e., ouabain and other digitalis drugs have been, and are being, used in cardiac failure and atrial fibrillation, while TBZ is currently used in the treatment of dystonic movements in Huntington’s disease (HD)). Facilitation of catecholamine release by these drugs may also distort the flight-or-fight response during stressful conflicts [8]. Eventually, this facilitation could find some therapeutic projection in dysautonomic diseases, certain types of shock or in some disorders that benefit from excessive dopamine release, such as the hyperactivity and attention deficit syndrome in children [9]. In this context, drugs that facilitate the release of adrenal catecholamines have interest from efficacy and safety points of view.

In our laboratory, we culminated the design and synthesis of new families of non-nucleotide purine derivatives. This program aimed at searching new blockers of the ionotropic purinergic receptor P2X7 (P2X7R). As this cation channel is emerging as a relevant gatekeeper of neuroinflammation [10], a P2X7R blocker could mitigate such process and displays activity in limiting the progression of neurodegenerative diseases; this was proven to be the case in the SOD1^G93A^ mouse model of amyotrophic lateral sclerosis (ALS) [11]. Among the newly synthesized compounds, we selected ITH15004 as a mild potent P2X7R blocker that mitigates the ATP-induced IL-1β release from macrophages and the [Ca^2+^]_c_ transients elicited by B_Z_ATP stimulation of human P2X7R-expressing HEK293 cells [12]. When exploring its general pharmacology, we found that compound ITH15004 (Figure 1) facilitated the release of catecholamines from isolated BCCs stimulated with K^+^ depolarizing pulses. Such facilitation seems to be due to alterations of Ca^2+^ handling by mitochondria, which in BCCs gives rise to a notable potentiation of exocytosis [13].

Here, we report the pharmacological effects of compound ITH15004 on catecholamine release and Ca^2+^ handling by BCCs. It seems that the compound may somehow interfere with mitochondrial Ca^2+^ movements to augment the [Ca^2+^]_c_ available for exocytosis during CC depolarization.

## 2. Results

### 2.1. Effects of Compound ITH15004 on Catecholamine Release

After placing a BCCs batch into the perfusion microchamber, real-time recording of catecholamine release was monitored at 37 °C for a 5 min period, in order to reach a stable baseline. Then, at 1 min intervals, cells were sequentially challenged with 5 s pulses of a high K^+^ solution (i.e., 35 mM K^+^, low Na^+^, the so-called 35K^+^ solution). Figure 2A shows an original recording of secretion responses elicited by each of the 35K^+^ pulses (P1 to P22) applied to each cell batch. In this experiment, the initial peak secretion amounted to 200 nA equivalents of catecholamines, that gradually decreased to reach 45.7% of P1 at P22. Pooled data from 14 cell batches from six different cell cultures show that the decay of exocytosis along the experiment reached 45.7 ± 2.2% of P1 at P22 (black square curve in Figure 2C).

In the experiment shown in Figure 2B, cells were similarly challenged with 35K^+^ pulses, displaying the typical decay of secretory responses during P1 to P6. However, upon cell exposure to 1 µM of ITH15004, the secretory peak responses underwent two notable changes: firstly, their gradual decay stopped; secondly, the responses gradually increased above the initial P1 value, to reach a plateau at P12, which slowly began to decay again upon compound washout. This behaviour is more clearly observed in the normalized averaged secretory responses (% P6) of the black circle curve in Figure 2C. The gradual augmented secretion is sustained in a plateau in the presence of ITH15004. Normalizing the response to P16 (the last 35K^+^ pulse in the presence of the compound), augmented secretion was estimated to be concentration-dependent at 0.3–1 µM; at higher concentrations (3–10 µM), facilitation of secretion was still present, but to a lesser extent. This suggests that two distinct mechanisms of action could be present at lower and higher ITH15004 concentrations, as argued in the Discussion.

### 2.2. Effects of ITH15004 on Secretion from Cells with Their Endoplasmic Reticulum Calcium Store Depleted

We have previously used a mixture of 10 mM caffeine, 1 µM ryanodine and 1 µM thapsigargin (the so-called CRT mixture) to deplete quickly, drastically, and irreversibly the ER Ca^2+^ store of CCs [14]. As ER Ca^2+^ mobilisation is involved in the potentiation by ouabain of secretion triggered by K^+^ in BCCs [6] and tetrabenazine [7], a possibility exists that ITH15004 was acting in a similar way. Thus, we tested the effects of the compound on secretion in BCCs with their ER Ca^2+^ store depleted.

Figure 3A shows an original trace from an experiment that started with six 35K^+^ control 5 s pulses. Subsequently, CRT was introduced in the perifusion fluid during the decay phase of P6, with an elevated baseline that was due to rapid ER Ca^2+^ mobilisation and depletion, and the stable subsequent lower spikes in the presence of CRT. Upon CRT washout the secretory spikes remained stable. In the original trace of Figure 3B, a similar protocol was used although ITH15004 (1 µM) was applied during the exposure of CCs to CRT. A progressive augmentation of secretion occurred. These responses decayed upon washout of both CRT and ITH15004. This increase of secretion, which is better observed in Figure 3C, where pooled results from eight experiments are plotted, was similar in the absence or the presence of CRT when secretion is normalized to control response in P16 (ellipse in panel C) secretion was 91.4 ± 6.2% in the presence of CRT and 149.4 ± 8.4% in the presence of CRT+ITH15004 (*p* < 0.001) (Figure 3D). In summary, facilitation of secretion by ITH15004 was still present in CCs with their ER Ca^2+^ store depleted and non-functional.

### 2.3. Effects of ITH15004 on Secretion upon the Pharmacological Interference with Mitocondrial Calcium Handling

Upon depolarization of BCCs, Ca^2+^ entry through VACCs creates a large submembrane Ca^2+^ transient that triggers exocytosis. Mitochondria contribute much to shape and clear such transient in these cells [13,15]. Mitochondria take up large amounts of Ca^2+^ through their Ca^2+^ uniporter (MICU), thanks to the existing large proton gradient [16,17,18]. Therefore, if such proton gradient is dissipated by a protonophore, mitochondria cannot contribute to clear up the large submembrane Ca^2+^ transient generated nearby subplasmalemmal exocytotic sites. Consequently, longer and larger Ca^2+^ transients occur, which give rise to higher exocytotic responses during K^+^ depolarization of BCCs [13,19]. Hence, experiments were done to find out whether compound ITH15004 modified somehow the effects of a protonophore on exocytosis.

Figure 4A shows experiments where a batch of BCCs were challenged with sequential 5 s pulses of 35K^+^ at 1 min intervals. The typical gradually declining secretory responses are observed during P1 to P6. At P7, the protonophore FCCP (3 µM) was introduced causing a drastic 3.5-fold potentiation of the 35K^+^ evoked response, which quickly returned to a mildly elevated baseline respect to the initial one. The response remained increased in P8 and then decayed progressively to reach a plateau at a mildly higher level than the pre-protonophore level of secretion. Then, secretion was further depressed upon FCCP washout at P16, but it started a slow recovery during P21–P22. In the experiment shown in Figure 4B, a similar protocol was followed. However, when the response was decaying in the presence of FCCP, ITH15004 at 1 µM was added. Secretion decay stopped, and the responses remained stable in a plateau from pulses P12 to P16, in which both compounds were present. Washout of both FCCP and ITH15004 resulted in a fast-brisk fall of secretion (P17 to P19), which started again to recover in the following 35K^+^ pulses. When added on top of FCCP, the effects of ITH15004 are better represented in the plot of pooled data shown in Figure 4C. For comparison, the time course of the decay of the secretion responses to repeated 35K^+^ challenging is also plotted (black squares). The decaying phase of the FCCP curve (black circles) is drastically stopped upon addition of ITH15004. The averaged normalized data on the effects of ITH15004 on secretion in the presence of FCCP, are represented in the bar graph of Figure 4D. At 0.3 µM, the compound did not modify the secretory response in the presence of FCCP, but at 1 and 3 µM ITH15004 increased such response by 72.9 ± 11.1% and 75.9 ± 10.6%, respectively. At 10 µM, however, the augmentation was milder (19.3 ± 5.6%).

We were interested in determining how ITH15004 affected the 35K^+^ responses when simultaneously added together with FCCP. Figure 5A represents an original recording showing a similar pronounced potentiation of secretion to that previously observed with FCCP alone (Figure 4A). However, an important difference with the experiment of Figure 4A arose: the amplitudes of the 35K^+^ secretory responses slightly decreased at P7 and P8 but were maintained high and non-inactivated in the subsequent P9 to P16 pulses. Upon washout of the two compounds, responses decayed drastically and quickly to very low levels, showing once more a tendency to recover in subsequent stimuli.

We also inquired whether the blockade of the mitochondrial Na^+^/Ca^2+^ exchanger (MNCX) with CGP37157 (1 µM) would affect the responses to 35K^+^ in the presence of combined FCCP+ITH15004. The simultaneous cell exposure to the three compounds produced the typical initial potentiation of secretion (P7 and P8). However, surprisingly, the presence of CGP37157 counteracted the effect of ITH15004 and provoked a quick decay of secretory responses to reach tiny amplitudes. Upon washout of the three compounds, the responses initiated a progressive recovery (Figure 5B).

Again, the time courses of the secretion responses are plotted in Figure 5C with pooled data. The decay of secretion in cells treated with the combined three compounds is consistently higher compared to the responses of cells exposed to only FCCP plus ITH15004. This was also the case for cell exposed to combined FCCP, ITH15004 and ruthenium red (RR, 1 µM), an MICU blocker. Plotting of the relative responses normalised to P16 (Figure 5D) shows the potentiation of secretion by FCCP plus ITH15004, and its drastic blockade by compound CGP37157 and by RR.

### 2.4. Effects of ITH15004 on Barium Currents

As K^+^-elicited secretion fully depends on Ca^2+^ entry through VACCs, we explored whether compound ITH15004 affected the whole-cell inward calcium currents triggered by depolarization of voltage-clamped BCCs. Facilitation of exocytosis by ITH15004 could be due to an augmented Ca^2+^ entry through VACCs, similarly to the dihydropyridine BayK8644 [20]. Figure 6A shows a family of current traces (2 mM Ba^2+^ as divalent cation carrier, I_Ba_), generated by sequential 50 ms depolarizing pulses to −10 mV, applied from a holding potential of −80 mV.

The initial peak current in each trace corresponds to inward Na^+^ currents (I_Na_), which is followed by the typical non-inactivating I_Ba_ (Control trace). This current (478 pA amplitude), was reduced to 392 pA in the presence of ITH15004 (1 µM) for 1 min. Added on top of the compound, ω-conotoxin GVIA (GVIA) at 1 µM, a N-type VACCs blocker further reduced I_Ba_ to 294 pA. Nifedipine at 10 µM, a L-type VACCs-blocking dihydropyridine, caused a further current blockade (to about 233 pA). Partial blockade of I_Ba_ corroborates the multiple VACC subtypes expressed by BCCs [4]. Washout of the compounds only partially recovered I_Ba_, as GVIA causes irreversible blockade of N-type VACCs [21].

Figure 6B better shows the time course of I_Ba_ blockade elicited by the cumulative exposure to the three compounds. The remaining unblocked I_Ba_ (around 50%) is due to the P/Q type of VACCs, that carries about half the total current in BBCs [4]. Pooled averaged data from 12 cells are plotted in Figure 6C. Relative blockade of I_Ba_ accounted for 19 ± 1.2%, 20.8 ± 0.8% and 17.9 ± 1.3%, for ITH15004, GVIA, and nifedipine, respectively.

### 2.5. Effects of ITH15004 on the Cytosolic Calcium Transients Elicited by K^+^

At each moment of cell activation, the size and shape of the [Ca^2+^]_c_ depend not only on Ca^2+^ entry, but also on its redistribution into the ER [22] and mitochondria [13]. Thus, we performed experiments to inquire how (if at all) compound ITH15004 could affect those intracellular Ca^2+^ movements. This was tested in cell populations of BCCs loaded with the Ca^2+^-sensitive fluorescent dye fluo-4-AM.

After recording for a minute to estimate basal [Ca^2+^]_c_ fluorescence, cells were challenged with 35K^+^ pulses (one pulse per well in 96-well plates). Changes of [Ca^2+^]_c_ were expressed as arbitrary fluorescence units (AFU) [7]. Figure 7A displays a family of [Ca^2+^]_c_ transient curves in the absence (control) and presence of increasing ITH15004 concentrations. At 0.3, 1, and 3 µM the compound augmented the [Ca^2+^]_c_ peaks elicited by 35K^+^. At 10 µM the [Ca^2+^]_c_ transient was also augmented but to a lesser extent. This is more clearly shown in Figure 7C where averaged data from 9 cell batches are plotted. The concentrations of 0.3, 1 and 3 µM augmented peak [Ca^2+^]_c_ by 71.6 ± 11.1%, 101.2 ± 9.8% and 87.6 ± 10.1%, respectively. At 10 µM, the [Ca^2+^]_c_ increase was 40.8 ± 22.8 but it did not reach the level of statistical significance when compared to control.

Of interest was the observation that baseline [Ca^2+^]_c_ was elevated by ITH15004, as shown in the original full traces shown in Figure 7B (arrow Basal). Dots represent 35K^+^ addition. Baseline elevations (recording of traces just before applying the 35K^+^ pulse) are averaged in Figure 7D. Basal fluorescence is significantly increased in cells pre-exposed to increasing concentrations of ITH15004. This augmented basal [Ca^2+^]_c_ is compatible with an effect of ITH15004 on Ca^2+^ mobilisation from an intracellular organelle. This was explored next.

### 2.6. Effects of ITH15004 on K^+^ Elicited [Ca^2+^]_c_ Elevations in Cells with their ER Calcium Store Depleted

The potential effect of ITH15004 on ER Ca^2+^ movements was explored in cells with their ER Ca^2+^ store irreversibly depleted by CRT. The effects of the compound on [Ca^2+^]_c_ were explored with experimental protocols like those presented in Figure 7.

As shown in Figure 8A, CRT itself augmented the [Ca^2+^]_c_ signals control response. Added on top of CRT, ITH15004 further augments the response (Figure 8A). Both CRT alone or CRT+ITH15004 at increasing concentrations augmented the [Ca^2+^]_c_ elevations elicited by 35K^+^ in a concentration-dependent manner, being maximal at 1–3 µM. Pooled data on the net increases of [Ca^2+^]_c_ normalized to% of control show that CRT itself caused a 43.7 ± 5.7% increase of the 35K^+^ elicited [Ca^2+^]_c_ transient. At 0.3 µM, ITH15004 added on top of CRT, did not produce any further [Ca^2+^]_c_ increase. However, 1 and 3 µM of ITH15004 caused a further 110.4 ± 14.7% and 113.8 ± 24.1% augmentation, respectively. Although to a lesser extent, 10 µM ITH15004 also caused an 82.4 ± 17.4% significant increase of [Ca^2+^]_c_ (Figure 8C)_._ Interestingly, CRT caused the expected elevation of basal Ca^2+^ (Figure 8A,D) and, moreover, addition of ITH15004 to CRT did not further increase basal [Ca^2+^]_c_ at all the concentrations (Figure 8D).

### 2.7. Effects of ITH15004 on 35K^+^ Elicited [Ca^2+^]_c_ Elevations upon Interfering with Mitochondrial Calcium Handling

In BCCs, mitochondria are a major contributor to the size and shaping of Ca^2+^ elevations elicited by 35K^+^ depolarization [13]. Thus, we explored whether ITH15004 affected the 35K^+^-evoked [Ca^2+^]_c_ elevations in fluo-4-loaded BCCs.

First, we tested whether protonophore FCCP at 1 µM, which dissipates the mitochondrial membrane potential and interrupt their ability to take up and retain Ca^2+^ from the cytosol, affected the elevations of [Ca^2+^]_c_ elicited by ITH15004. Figure 9A shows that FCCP augmented the 35K^+^ induced [Ca^2+^]_c_ transients, as expected. Added on top of FCCP, ITH15004 still caused further elevations of [Ca^2+^]_c_ in a concentration-dependent manner (Figure 9A,B).Added FCCP in combination with the MICU blocker RR at 1 µM, the [Ca^2+^]_c_ elevation was 107% (a 42% higher than when FCCP was added alone; comparing panels B and D of Figure 9). Under these conditions, ITH15004 added on top of FCCP+RR did not increase further the 35K^+^-elicited [Ca^2+^]_c_ transient (Figure 9C,D). This indicates that MICU seems to be involved in the augmentation of [Ca^2+^]_c_ elevations elicited by ITH15004.

Another experiment explored the possible involvement of MNCX in the effects of ITH15004 on [Ca^2+^]_c_. Thus, 1 µM of the MNCX blocker CGP37157 [13] was used in combination with FCCP. Combined FCCP+CGP elicited a 60% increase of [Ca^2+^]_c_, a lesser potentiation compared with the augmentation produced by FCCP+RR (comparing panels D and F of Figure 9). Added on top of FCCP+CGP, ITH15004 did not increase further the 35K^+^-evoked [Ca^2+^]_c_ elevations (Figure 9E,F).

Finally, we investigated whether compound JNJ-47965567 (JNJ), a potent blocker of P2X7Rs, was mimicking somehow the effects of ITH15004 on mitochondrial Ca^2+^ handling. Combined FCCP+JNJ augmented by 53% the [Ca^2+^]_c_. Added on top of FCCP+JNJ, compound ITH15004 augmented further the [Ca^2+^]_c_ responses in a concentration-dependent manner. Note that panels B (FCP+ITH15004) and H (FCCP+JNJ+ITH15004) show quite similar effects on [Ca^2+^]_c_ elevations, suggesting that P2X7Rs are not involved in the effects of compound ITH15004 on the [Ca^2+^]_c_ transients elicited by 35K^+^ in BCCs. Additionally, JNJ did not affect the facilitation of secretion elicited by 35K^+^ (data not shown).

## 3. Discussion

The central finding of this investigation is the facilitation of the K^+^-elicited exocytotic release of catecholamines from BCCs by the novel purine derivative compound ITH15004 (Figure 2B). As the K^+^ secretory response strictly depends on Ca^2+^ ions [23], it seems plausible to link those effects to some influence of ITH15004 on the handling of Ca^2+^ by BCCs. The secretory responses of BCCs upon their challenging with sequential 35K^+^ pulses applied at regular intervals underwent a progressive decline (Figure 2A). Decay of spike amplitudes may have various components affecting either Ca^2+^ handling by the cell and/or vesicle transport and priming after each 35K^+^ pulse, whereby the secretory machinery is depleted of a ready-release vesicle pool [3]. In BCCs, we have previously suggested that the lower [Ca^2+^]_c_ elevations required for vesicle transport and priming [24] is controlled by ER Ca^2+^ fluxes [22,25] and/or by mitochondrial Ca^2+^ movements [13,26]. As depletion of ER Ca^2+^ with CRT did not affect the ability of ITH15004 to augment exocytosis (Figure 3B), it seemed that this Ca^2+^ store is not contributing to the effects of the compound. This was proven by an experiment monitoring the [Ca^2+^]_c_ transients elicited by K^+^; upon ER Ca^2+^ depletion with CRT, cells still responded with higher [Ca^2+^]_c_ transients in the presence of increasing concentrations of ITH15004 (Figure 8B).

In sharp contrast, ITH15004 deeply affected the secretory responses under conditions of pharmacological manipulation of Ca^2+^ handling by mitochondria. The drastic potentiation of secretion by FCCP is due to impaired Ca^2+^ uptake (and hence of Ca^2+^ circulation) in mitochondria [13]. The decay of those responses in the continued presence of FCCP (Figure 4A) are likely dependent on mitochondrial deterioration of Ca^2+^-dependent ATP production [17,18] that is necessary to maintain exocytosis [3]. Compound ITH15004 exhibited two effects on these FCCP-potentiated secretory responses, namely slowing down of secretory decay (when added after FCCP) and full prevention of secretion decay (when added with FCCP). This strongly supports the view that ITH15004 is facilitating exocytosis by acting on Ca^2+^ handling by mitochondria. This was proven through the monitoring of [Ca^2+^]_c_ elevations of basal [Ca^2+^]_c_ and [Ca^2+^]_c_ elevations triggered by 35K^+^. While ITH15004 did not exert major changes of [Ca^2+^]_c_ under conditions of ER Ca^2+^ depletion, its effects on mitochondrial Ca^2+^ were notorious. Thus, under cell exposure to FCCP when mitochondria are unable to trap cytosolic Ca^2+^ because of the dissipation of the proton gradient and mitochondrial depolarization [17,18], ITH15004 was still capable of increasing the 35K^+^-induced [Ca^2+^]_c_ elevations, even above the levels previously augmented by FCCP alone. Furthermore, additional observations showed that the interruption of mitochondrial Ca^2+^ circulation, either by RR (blockade of Ca^2+^ influx) or CGP (blockade of Ca^2+^ efflux), cancelled the augmentation of [Ca^2+^]_c_ elicited by ITH15004 in the presence of FCCP. This suggests two interesting mechanisms involved in ITH15004 effects. First, the compound requires active mitochondrial Ca^2+^ circulation even in the presence of FCCP and is reinforced by ITH15004, that might contribute to maintain ATP synthesis. In so doing, it is plausible that ITH15004 is facilitating on the one hand, vesicle transport and priming of the secretory machinery by maintaining for longer time the elevated basal [Ca^2+^]_c_ during the resting periods in between the repeated 35K^+^ pulses. On the other hand, through its effects in facilitating mitochondrial Ca^2+^ circulation, ITH15004 may contribute to maintain ATP synthesis and hence, the ATP required for maintaining healthy secretory responses upon repeated stimuli of BCCs. How these effects are exactly exerted, and which are the mitochondrial targets on which ITH15004 is acting, is a matter of further investigation.

A last point deserves attention. As explained in the Introduction, compound ITH15004 was developed in the frame of a program searching for new purine derivatives with potential ability to block P2X7Rs. As CCs are known to express various subtypes of purinergic receptors [27], the possibility exists that the effects here described for ITH15004, a mildly potent P2X7R blocker with an IC_50_ of 14 µM [12], could be mediated by those receptors. This did not seem to be the case for two reasons: (i) the effects here described on Ca^2+^ handling and exocytosis, were maximally observed at 1–3 µM, concentrations that do not target P2X7Rs [12]; and (ii) potent P2X7R blocker JNJ47965567 (with a nanomolar-ranged IC_50_), did not affect either Ca^2+^ handling or exocytosis.

In conclusion, compound ITH15004 elicited both a mild potentiation of catecholamine secretory responses triggered by BCC depolarization and a maintenance of those enhanced responses for longer time. These effects are exerted by interfering with Ca^2+^ handling by mitochondria. If this improvement is also exerted under mitochondrial deficits of Ca^2+^ handling and ATP synthesis, compound ITH15004 may become a useful pharmacological tool to correct those deficits in, for instance, cell, tissue, and animal models of neurodegenerative diseases. In the light of the results here obtained, further investigation on cell and animal models of those diseases are warranted. Additionally, the precise target(s) that regulate mitochondrial Ca^2+^ circulation, on which ITH15004 could act, is a subject worth of further investigation.

## 4. Materials and Methods

### 4.1. Preparation of Bovine Chromaffin Cell Cultures

BCCs were isolated from adrenal glands of adult cows following standard methods [28] with some modifications [29,30]. They were plated at a density of 5 × 10^6^ cells per 5 mL of Dulbecco’s modified Eagle’s medium (DMEM) in 6 cm diameter Petri dishes (to study secretion), in 12 mm-diameter glass coverslips at a density of 5 × 10^4^ cells (to study barium currents, I_Ba_), or into 96-well black plates at a density of 2 × 10^5^ cells per well, to monitor Ca^2+^ signals. Experiments were performed 2–4 days after plating.

### 4.2. Real-Time Monitoring of Catecholamine Release Triggered by High-K+ Depolarizing Pulses

Cells were scrapped off carefully from the bottom of the Petri dish (5 × 10^6^ cells per dish) with a rubber policeman and centrifuged at 100x *g* for 10 min. The cell pellet was resuspended in 200 µL of Krebs-HEPES solution (composition in mM: 144 NaCl, 5.9 KCl, 1.2 MgCl_2_, 11 glucose, 10 HEPES, 2 CaCl_2_, pH 7.4 adjusted with NaOH). Cells were introduced in a 100 µL microchamber, previously filled with a filter for cell retention, and perfused at a rate of 2 mL/min. The liquid flowing from the superfusion chamber reached an electrochemical detector (model CH-9100; Metrohm AG, Herisau, Switzerland) placed just at the outlet of the microchamber, which monitors in real time the amount of catecholamines secreted under the amperometry mode. The catecholamines were oxidized at +0.65 V and the oxidation current was recorded with a frequency of 2 Hz, to monitor the amount of total catecholamine secreted [31]. To stimulate the catecholamine secretion, a Krebs–Hepes solution containing KCl at 35 mM with isosmotic reduction of NaCl (35K^+^ solution), was applied in 5 s pulses at 1 min regular intervals at 37 ºC. After each stimulation, we measured in real time the catecholamine release by amperometry [30,31,32]. Representative records of some experiments shown in this report were accomplished by importing the data obtained in ASCII format to the Origin 8.6 (Microcal) program.

### 4.3. Electrophysiological Recording

Barium currents were recorded using the whole-cell configuration of the patch-clamp technique [33]. Coverslips containing the cells were placed on an experimental chamber mounted on the stage of Nikon Diaphot inverted microscope. Cells were perifused at 22 ± 2 °C with a Tyrode solution containing (in mM): 137 NaCl, 1 MgCl_2_, 2 CaCl_2_, 10 HEPES, pH 7.4 (adjusted with NaOH). Once the patch membrane was ruptured and the whole cell configuration of the patch-clamp technique was established, the cell was locally, rapidly and constantly perfused with an extracellular solution of similar composition to the chamber solution, but containing 2 mM Ba^2+^ to monitor I_Ba_. In this way, we obtained greater and less inactivating currents than if we used 2 mM of Ca^2+^. Moreover, 2 mM of Ba^2+^ allows us to see the maximum effect of toxins in blocking the different VACC subtypes in CCs [34]. Cells were internally dialyzed with an intracellular solution containing (in mM): 100 Cs-glutamate, 14 EGTA, 20 TEA·Cl, 10 NaCl, 5 Mg·ATP, 0.3 Na·GTP, and 20 HEPES, pH 7.3 (adjusted with CsOH). For I_Ba_ data acquisition, cells were voltage-clamped at −80 mV and step-depolarisations with 50 ms depolarising pulses were applied at 30-s intervals to minimise current rundown [35]. Data were acquired with a sampling frequency of 20 kHz using the PULSE 8.74 software (Heka Elektronic, Lambrecht, Germany). Linear leak and capacitive components were subtracted by using a P/4 protocol and series resistance was compensated by 80%. The data analysis was performed with Igor Pro (Wavemetrics, Lake Oswego, OR, USA) and PULSE programs (Heka Elektronik).

### 4.4. Monitoring of Cytosolic Calcium Levels

To monitor the changes of [Ca_2+_]_c_, cells were plated at a density of 2 × 10^5^ cells per well into 96-well black plates, and the experiments were performed 48 h later. Cells were loaded with a Krebs-HEPES solution (composition in mM: 144 NaCl, 5.9 KCl, 1.2 MgCl_2_, 11 glucose, 10 HEPES, 2 CaCl_2_, pH 7.4 adjusted with NaOH) with 10 µM fluo-4-AM and 0.2% pluronic acid. Cells were incubated for 45 min at 37 °C in the dark. After this incubation period, cells were washed twice with the Krebs-HEPES solution at room temperature in the dark. Changes in fluorescence (excitation 485 nm, emission 520 nm) were measured using a fluorescent plate reader (Fluostar; BMG Labtech, Offenburg, Germany). Basal fluorescence levels were monitored before adding the stimulation solution (35K+) with an automatic dispenser. After stimulation of the cells, changes in fluorescence were measured for 60 s. To normalize fluo-4 signals, responses from each well were calibrated by measuring maximum and minimum fluorescence values. At the end of each experiment, 5% Triton X-100 (F_max_) was added, followed by 2 mM MnCl_2_ (F_min_). Data were calculated as a percentage of F_max_–F_min_.

### 4.5. Chemicals

The following chemicals were used: collagenase type I, FCCP (carbonyl cyanide 4-(trifluoromethoxy)phenylhydrazone), ruthenium red (RR), caffeine, ryanodine (Ry), thapsigargin, and CGP37157, were from Sigma/MercK (Madrid, Spain); Dulbecco’s modified Eagle’s medium (DMEM), bovine serum albumin fraction V, foetal calf serum, and antibiotics were from Gibco (Madrid, Spain). Fluo-4-AM from Life Technologies (Madrid, Spain). JNJ-47965567 was provided by Janssen Pharmaceutica NV (Beerse, Belgium). All the other chemicals used were reagent grade from Merck and Panreac Química (Madrid, Spain). FCCP, RR, Ry and Thapsigargin mother solutions (10^−2^ M) were prepared in dimethyl sulfoxide (DMSO) and protected from light. Final drug concentrations were obtained by diluting the stock solutions directly into the extracellular solution. At these dilutions, solvents had no effect on the parameters studied.

### 4.6. Data Analysis

Data are presented as single examples to illustrate the experimental protocols and as averaged pooled data from several experiments, as means ± SEM of the number of different experiments done (number of cells, n) and different cultures used (N). Mathematical analyses were done using the GraphPad Prism software, version 5.01 (GraphPad Software Inc., La Jolla, CA, USA). Comparison between means of group data were performed by one-way analysis of variance (ANOVA) following by the Tukey’s post hoc test when appropriate.

## Figures and Tables

**Figure 1 ijms-23-00440-f001:**
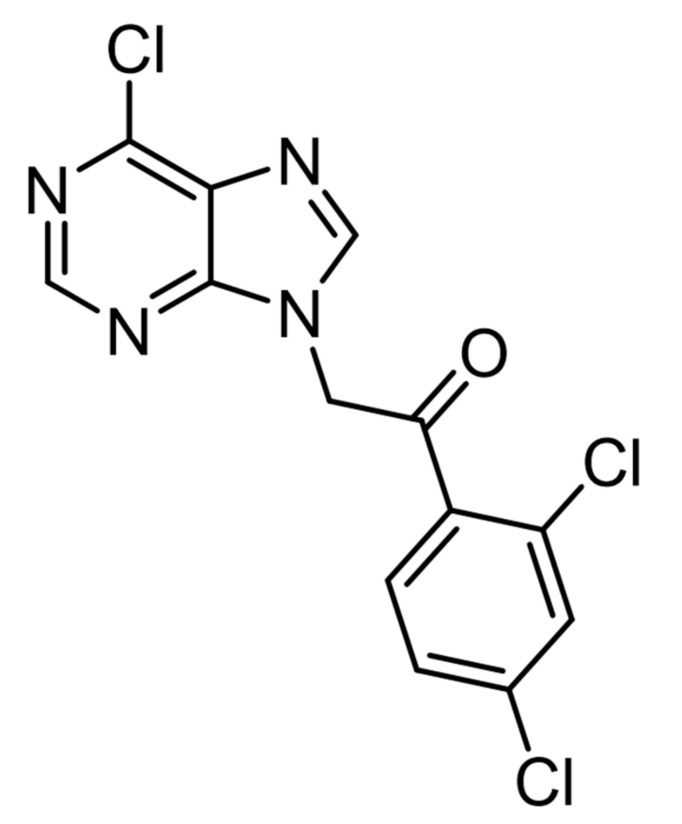
Chemical formula of compound ITH15004: [2-(6-chloro-9*H*-purin-9-yl)-1-(2, 4-dichlorophenyl) ethenone].

**Figure 2 ijms-23-00440-f002:**
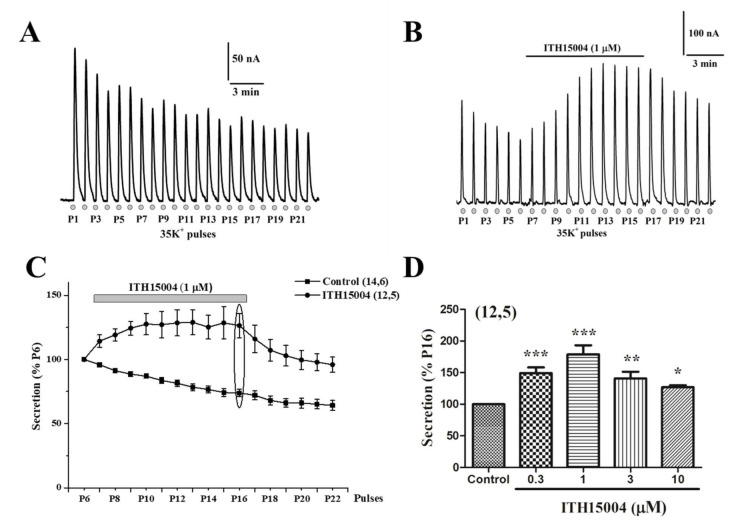
Facilitation of exocytosis by compound ITH15004. The exocytotic release of catecholamines was triggered by sequential stimulation of fast-perifused bovine chromaffin cells (BCCs) with a 35 mM K^+^, low Na^+^ solution (35K^+^) during 5 s at 3 min intervals (bottom dots in panels (**A**) and (**B**), 22 pulses were applied, P1 to P22). (**A**), original record of secretory responses, showing a gradual decay in control cells. (**B**), original record of secretory responses elicited by 35K^+^ challenges in cells that were exposed to compound ITH15004 during the time period indicated by the top horizontal bar. Insets in A and B are calibration bars (secretion of catecholamines in nA versus time). (**C**), normalized averaged secretion expressed as percentage of P6 in each individual experiment (ordinate), expressing the time course (abscissa) of secretory spike amplitude in control cells and cells exposed to ITH15004 (top horizontal bar, pulses P7 to P16). Data are means ± SEM of the number of experiments (n) and the different cell cultures (N) shown in parentheses (n,N). (**D**), pooled data on the concentration-response effects of ITH15004 in enhancing the secretory responses to 35K^+^ pulses. Data are normalized as percentage of P16 and correspond to the last pulse P16 in both control cells (100% ordinate) and cells exposed to increasing concentrations of ITH15004 (abscissa) (points marked with an ellipse in panel (**C**)). They are means ± SEM of the number of experiments (n) and separate cell cultures (N) shown in parenthesis (n,N). * *p* < 0.05, ** *p* < 0.01, *** *p* < 0.001 with respect to control.

**Figure 3 ijms-23-00440-f003:**
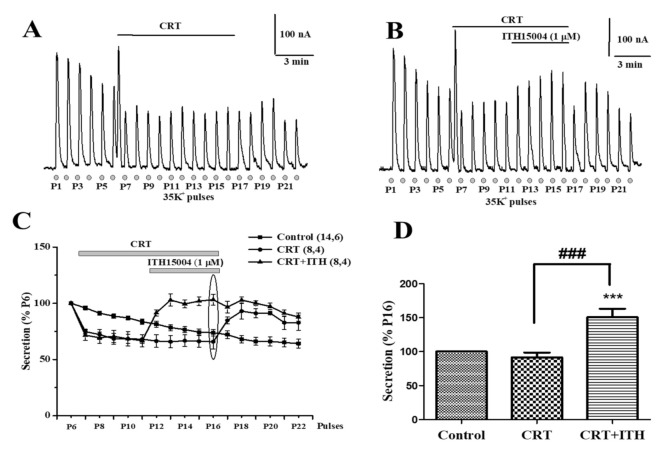
Compound ITH15004 still augments the secretory responses elicited by 35K^+^, in BCCs with their endoplasmic reticulum (ER) calcium store depleted. ER depletion is caused by cell exposure to the CRT mixture (20 mM caffeine, 1 µM ryanodine, 1 µM thapsigargin). (**A**), original record of secretory responses before, during CRT exposure and after its washout. (**B**), original record of secretory responses before, during CRT or CRT+ITH15004 cell exposure, as indicated by top horizontal lines. Insets, calibration bars. (**C**), averaged quantitative data of secretory responses normalized as percentage of P6 in each individual experiment. (**D**), secretory responses normalized to P16 (ellipse in panel (**C**)), in the presence of CRT alone or combined CRT plus ITH15004. Data are means ± SEM of the number of experiments (n) and distinct cell cultures (N) shown in parenthesis (n,N). *** *p* < 0.001 with respect to control; ^###^ with respect CRT.

**Figure 4 ijms-23-00440-f004:**
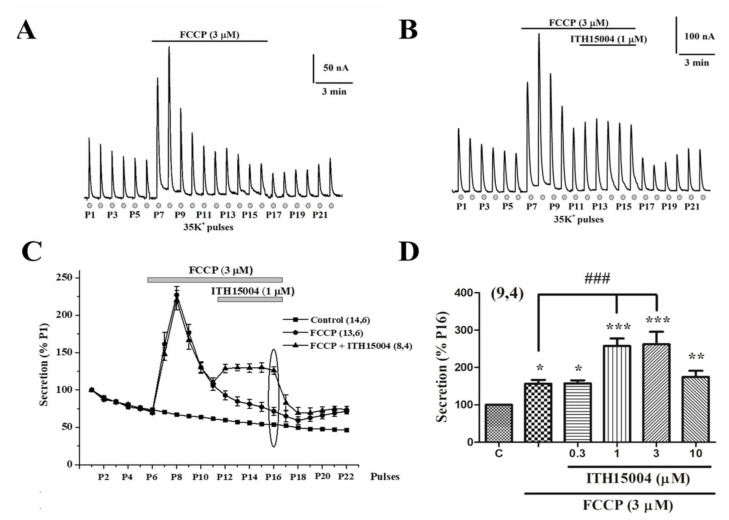
Effects of ITH15004 on the secretory responses elicited by 35K^+^ pulses, in cells exposed to the protonophore FCCP. (**A**), original record of secretion responses before, during FCCP cell exposure (top horizontal line) and after its washout. Inset, calibration bars (secretion in nA and time). (**B**), original record of secretory responses before and during cell exposure to FCCP (top horizontal line), and during cell exposure to ITH15004 added on top of FCCP during the time period indicated by the horizontal bar. Inset, calibration bars on secretion in nA and time in min. (**C**), pooled data on the time course of secretory response elicited by 35K^+^ pulses in control conditions (bottom curve) and during cell exposure to FCCP alone or to combined FCCP plus ITH15004, according to the time periods marked by the top horizontal lines. Data are means ± SEM of the number of experiments (n) and different cell cultures (N) shown in parenthesis. (**D**), pooled data on the experiments shown in panel C at P16 (ellipse); they represent the effects of FCCP alone and of FCCP combined with increasing concentrations of ITH15004 (bottom horizonal lines). Data are means ± SEM of the number of experiments (n) and separate cell cultures (N) shown in parenthesis (n,N). * *p* < 0.05, ** *p* < 0.01, *** *p* < 0.001 with respect control. ^###^
*p* < 0.001 with respect to FCCP alone.

**Figure 5 ijms-23-00440-f005:**
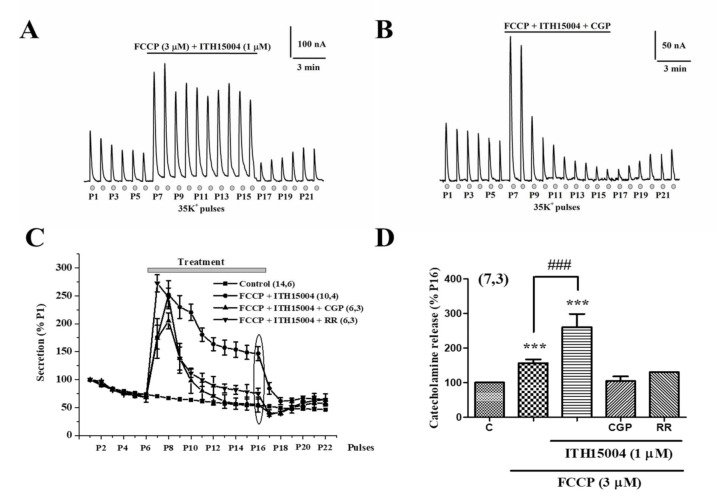
Effects of simultaneous cell exposure to FCCP+ITH15004 on secretory responses triggered by 35K^+^ in BCCs, upon the pharmacological manipulation of calcium handling bye mitochondria. (**A**), original record on secretory responses before and during cell exposure to FCCP+ITH15004, during the time period marked by the horizontal top line. Inset, calibration bars of spike secretion amplitude in nA and time in minutes. (**B**), original record on secretory responses before and during cell exposure to combined FCCP+ITH15004 + CGP37157. (**C**), averaged data normalized to P1 (ordinates), on the time course of secretory responses in control cells (bottom curves) and cells exposed to combined FCCP+ITH15004, to FCCP+ITH15004+CGP37157 or to combined FCCP, ITH15004 and ruthenium red (1 μM) (top bar on treatments). Data are means ± SEM of the number of experiments (n) and different cell cultures (N) shown in parentheses. (**D**), pooled data normalized as percentage of P16 (as indicated by the ellipse in panel C) (ordinate) on the effects of the different treatments namely, FCCP alone, FCCP+ITH15004 (ITH), FCCP+ITH15004+CGP37157 at 1 μM (CGP) and FCCP+ITH15004+ ruthenium red (RR). Data are means ± SEM of the number of experiments (n) done in different cultures (N) shown in parenthesis (n,N). *** *p* < 0.001 with respect control (C). ^###^
*p* < 0.001 with respect to FCCP alone.

**Figure 6 ijms-23-00440-f006:**
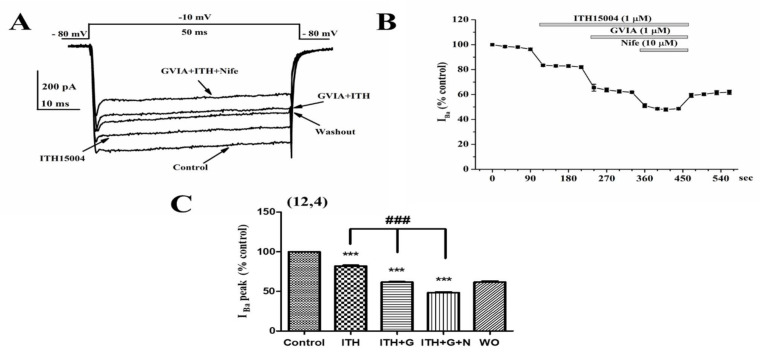
ITH15004 (ITH) mildly blocks the whole-cell current through VACCs in voltage-clamped BCCs using 2 mM external Ba^2+^ as charge carrier. The protocol shown on top of panel A was used to elicit I_Ba_ by sequential 50 ms pulses to −10 mV (peak current) from a holding potential of −80 mV. (**A**), family of original Ba^2+^ current records obtained in control conditions (note the initial peak in each trace, corresponding to sodium current, I_Na_), and the traces obtained after 1 min exposure of the cell to 1 µM ITH15004 (ITH), combined ITH+1 µM ω-conotoxin GVIA (GVIA), or combined ITH+GVIA+10 µM nifedipine (Nife). Inset, calibration bars of currents (pA) and time (ms). (**B**), time course of the normalized I_Ba_ (initial current amplitude = 100%, ordinate) and its blockade by cumulative additions of ITH15004, GVIA and Nife, as indicated by the top horizontal bars. Abscissa, time in seconds. (**C**), pooled data on the blockade of I_Ba_ (normalized as percentage of the initial I_Ba_ peak, ordinate) elicited by cumulative addition of ITH, then GVIA (G) and finally nifedipine (N). WO, washout. Data are means ± SEM of the number of cells tested (n) from different cell cultures (N) shown in parenthesis (n,N). *** *p* < 0.001 with respect control. ^###^
*p* < 0.001 with respect to ITH15004 alone.

**Figure 7 ijms-23-00440-f007:**
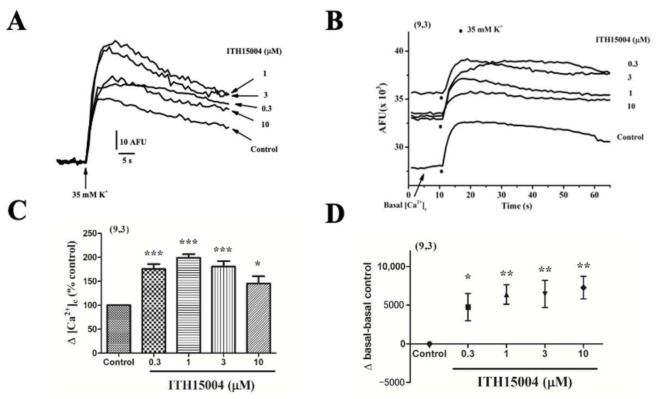
ITH15004 augments the cytosolic calcium signals ([Ca^2+^]_c_) elicited by 35K^+^ pulses applied to fluo-4-loaded BCCs. Cells seeded in 96 well black plates were exposed to increasing concentrations of ITH15004 10 s before their stimulation with a pulse of 35K^+^. (**A**), family of [Ca^2+^]_c_ traces normalized to the initial baseline in the absence (Control) and the presence of the indicated concentrations of ITH15004. Inset, calibration bars in net increases of arbitrary fluorescence units (AFU), versus time (s). (**B**), [Ca^2+^]_c_ signals elicited by 35K^+^, in absolute AFU (ordinate) versus time (abscissa), before (control) of after cell exposure to increasing concentrations of ITH15004. Note the initial basal [Ca^2+^]_c_ in each trace that is augmented in the presence of ITH15004 (curves are averaged from 9 experiments done in 3 different cultures). (**C**), quantitative averaged data on the net [Ca^2+^]_c_ increase elicited by 35K^+^ in the absence (Control) and the presence of ITH15004 (bottom horizontal line). (**D**), net baseline increases (after subtracting baseline control) (ordinate) elicited by ITH15004 (abscissa). Data in panels C and D are means ± SEM of 9 experiments from 3 different cell cultures, as indicated in parenthesis * *p* < 0.05, ** *p* < 0.01, *** *p* < 0.001, with respect to control.

**Figure 8 ijms-23-00440-f008:**
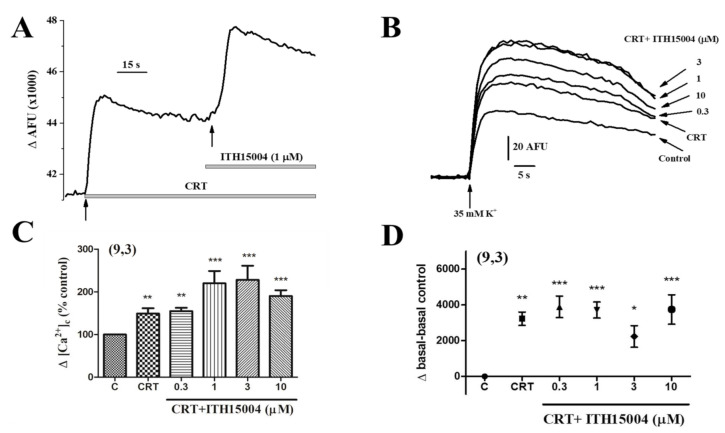
Augmentation by ITH15004 of the [Ca^2+^]_c_ signals elicited by 35K^+^, in BCCs with their ER Ca^2+^ store depleted by CRT (a mixture of 20 mM caffeine, 1 µM ryanodine, and 1 µM thapsigargin). (**A**), CRT itself augmented the [Ca^2+^]_c_ signals control response. Added on top of CRT, ITH15004 further augments the response. (**B**), ITH15004 augments the [Ca^2+^]_c_ elevations elicited by 35K^+^ in cells treated with CRT, in a concentration-dependent manner. (**C**), pooled data on the net increases of [Ca^2+^]_c_ normalized to% of control in each individual experiment (ordinate), obtained from experiments of panel B. (**D**), pooled data on baseline elevations, expressed in absolute AFU (ordinate). Data in C and D are means of 9 experiments from 3 different cultures, as indicated in parenthesis. * *p* < 0.05, ** *p* < 0.01, *** *p* < 0.001 with respect to control.

**Figure 9 ijms-23-00440-f009:**
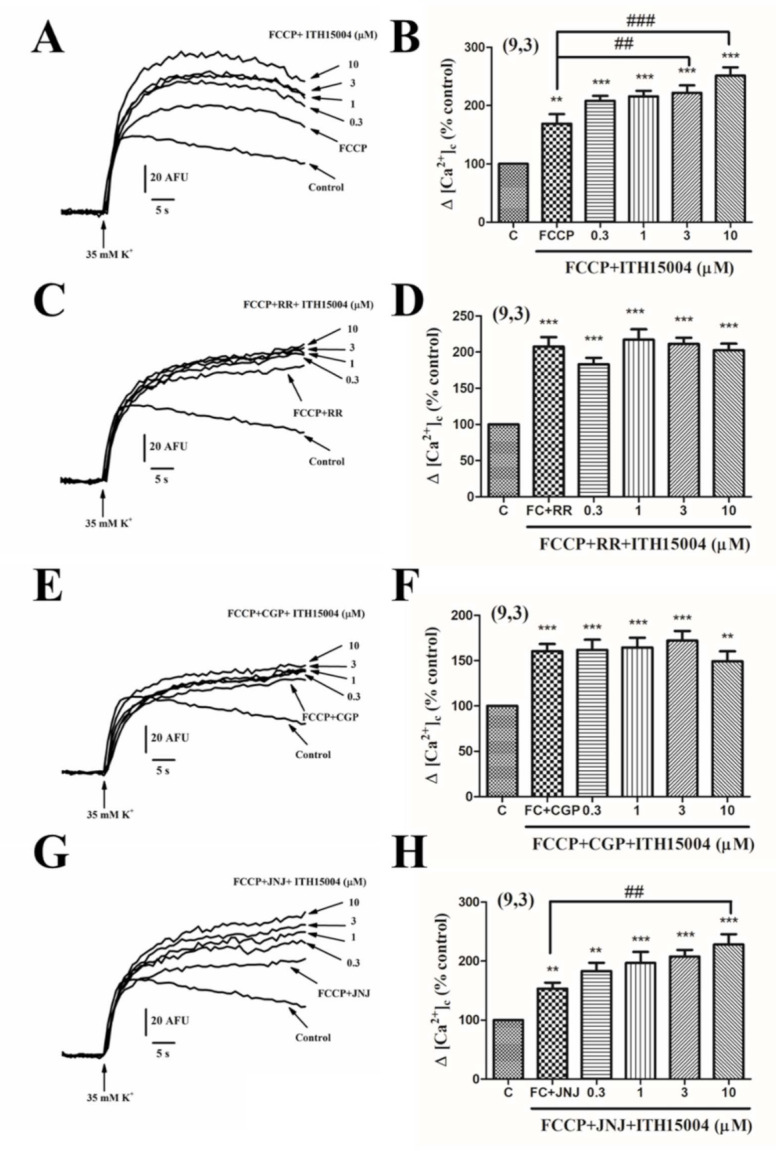
Effects of ITH15004 combined with FCCP on 35K^+^ elicited [Ca^2+^]_c_ elevations and the effects of RR 1 µM or CGP371757 (CGP, 3 µM). (**A**,**B**) original traces and pooled data on the effects of FCCP alone and FCCP plus increasing concentrations of ITH15004, respectively, on the [Ca^2+^]_c_ transients elicited by 35K^+^. (**C**,**D**) similar experiments but with a combination of FCCP+RR+increasing concentrations of ITH15004. (**E**,**F**), similar experiments (as in (**A**,**B**)) but with a combination of FCCP (FC)+CGP plus increasing concentrations of ITH15004. (**G**,**H**), similar experiments with combined FCCP (FC)+JNJ-47965567+increasing concentrations of ITH15004. Pooled data in panels B, D, F, H are means ± SEM of 9 experiments from 3 different cell cultures, as indicated in parenthesis. ** *p* < 0.01, *** *p* < 0.001 with respect control (**C**). Panels B and H; ^##^
*p* < 0.01, ^###^
*p* < 0.001 respect to FCCP (**B**) or FCCP+JNJ (**H**).

## Data Availability

The data presented in this study are available on request from the corresponding author. The data are not publicly available due to privacy.

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
