# Peer review of "Novel Purine Derivative ITH15004 Facilitates Exocytosis through a Mitochondrial Calcium-Mediated Mechanism"

_ijms, 2021, doi:10.3390/ijms23010440_

Round 1
Reviewer 1 Report
Authors show a great continuation of their previous work (Francesco Calzaferri, Paloma Narros-Fernández, Ricardo de Pascual, Antonio M G de Diego, Annette Nicke, Javier Egea, Antonio G García, Cristóbal de Los Ríos. Synthesis and Pharmacological Evaluation of Novel Non-nucleotide Purine Derivatives as P2X7 Antagonists for the Treatment of Neuroinflammation. J Med Chem. 2021;64(4):2272-2290. doi: 10.1021/acs.jmedchem.0c02145). In this multi step study they found that novel purinergic receptor antagonist facilitates catecholamines release through calcium dependent mechanism. I have a question to Authors, do you see any potential threats theoretically related with the release of catecholamines or stimulation of calcium-related mechanisms? Should we be scared of calcium dependent neurotoxicity?
Author Response
Dear Reviewer, thank you very much for your comments. We hope our responses were worth to consider our manuscript for publication in IJMS.
Reviewer: "Authors show a great continuation of their previous work (Francesco Calzaferri, Paloma Narros-Fernández, Ricardo de Pascual, Antonio M G de Diego, Annette Nicke, Javier Egea, Antonio G García, Cristóbal de Los Ríos. Synthesis and Pharmacological Evaluation of Novel Non-nucleotide Purine Derivatives as P2X7 Antagonists for the Treatment of Neuroinflammation. J Med Chem. 2021;64(4):2272-2290. doi: 10.1021/acs.jmedchem.0c02145). In this multi step study they found that novel purinergic receptor antagonist facilitates catecholamines release through calcium dependent mechanism.
I have a question to Authors, do you see any potential threats theoretically related with the release of catecholamines or stimulation of calcium-related mechanisms?"
RESPONSE: Thank you very much. We do not think that the release of catecholamines that could be directly induced by ITH15004 can be a threat and/or be able to induce adverse cardiovascular effects related to an excess of catecholamines (e.g. hypertension, cardiac arrhythmias). On the opposite, as indicated, the main finding of our manuscript is that ITH15004, in addition to its blocking effect of purinergic P2X7 receptors (thus protecting against neuroinflammation in neurodegenerative diseases), is also acting on the management/regulation of Ca2+ by mitochondria during cell activation. Hence, based on this aditional mechanism of action, ITH15004 could constitute a new pharmacological tool to study the possible alterations both in mitochondrial bioenergetic and its role in the regulation of calcium handling related to neurotransmitters exocytosis in excitable cells, which are two phenomena that are also altered during development and progression of neurodegenerative diseases.
REVIEWER: "Should we be scared of calcium dependent neurotoxicity?"
RESPONSE: As for neurotransmitter exocytosis, regulation of mitochondrial calcium handling has been described as a neuroprotective mechanism and, therefore, we think that the effect of ITH15004 on mitochondria-regulated calcium fluxes might contribute to its neuroprotective effects mediated by its main mechanism of action related to the blockade of P2X7 receptors.
We thank your comments and suggestions on our manuscript, that help to improved our Ms.
Reviewer 2 Report
The manuscript: „Novel purine derivative ITH15004 facilitates exocytosis through a mitochondrial calcium-mediated mechanism" by Ricardo de Pascual and colleagues describe a novel pharmacological tool that is helpful in studying the role of mitochondria in the regulation of the bioenergetics and exocytosis in excitable cell. The manuscript is nicely composed with promising methodology and convincing results.
After thoroughly going through the manuscript, I have a couple of comments:
- ITH15004 is already known to be the most potent, selective, and blood-brain barrier-permeable antagonist and its role as a first non-nucleotide purine in in future therapeutic aspects in neuroinflammation has been intensively studied at the moment. Please briefly describe this aspect based on your results.
- Was sample size analysis performed? Had the sample size sufficient statistical power for the subsequent statistical analyses performed?
Author Response
Dear Reviewer, thank you very much for your comments. Please find our responses below.
REVIEWER: The manuscript: "Novel purine derivative ITH15004 facilitates exocytosis through a mitochondrial calcium-mediated mechanism" by Ricardo de Pascual and colleagues describe a novel pharmacological tool that is helpful in studying the role of mitochondria in the regulation of the bioenergetics and exocytosis in excitable cell. The manuscript is nicely composed with promising methodology and convincing results. After thoroughly going through the manuscript, I have a couple of comments:
- ITH15004 is already known to be the most potent, selective, and blood-brain barrier-permeable antagonist and its role as a first non-nucleotide purine in in future therapeutic aspects in neuroinflammation has been intensively studied at the moment. Please briefly describe this aspect based on your results.
RESPONSE: Thank you very much, our current results describe a “secondary” mechanism of action that could serve to improve the expected neuroprotective effects of ITH15004 mediated through the blockade of P2X7 receptors implicated in neuroinflammation processes in many neurodegenerative diseases. As described in our manuscript, in addition to these effects, this compound could serve to modulate mitochondrial calcium handling and to restore mitochondrial bioenergetics that are also altered in neurodegenerative diseases. Chromaffin cells and catecholamine secretion are widely used as an experimental model to study the contribution of mitochondria and/or the endoplasmic reticulum to the regulation of intracellular calcium handling, and its disturbances in neurodegenerative diseases.
REVIEWER:
- Was sample size analysis performed? Had the sample size sufficient statistical power for the subsequent statistical analyses performed?
RESPONSE: In the experiments included in this manuscript, we have the advantage of working with cell populations (5 million cells per experiment) which significantly reduces the possible biological variability between experiments and allows us to obtain statistically different results with a minimum of 6 independent experiments, carried out with cells from at least 2 different cell cultures. Otherwise, in our experiments, the minimum number of cell cultures that we have used is 4, and the minimum number of independent experiments is 8, as indicated in the corresponding figures. As indicated in the manuscript, comparison between means of group data were performed by one-way analysis of variance (ANOVA), obtaining the levels of statistical significance indicated in each figure.
We thank your comments and suggestions on our manuscript, that help to improved our Ms.